# Reciprocal regulation of STING and TCR signaling by mTORC1 for T-cell activation and function

Takayuki Imanishi[1], Midori Unno[1], Wakana Kobayashi[1], Natsumi Yoneda[1], Satoshi Matsuda[2], Kazutaka Ikeda[3,4,5], Takayuki Hoshii[6], Atsushi Hirao[6], Kensuke Miyake[7], Glen N Barber[8], Makoto Arita[3,5,9], Ken J Ishii[10,11], Shizuo Akira[12], Takashi Saito[1,13]

**Stimulator of interferon genes (STING) plays a key role in detecting cytosolic DNA and induces type I interferon (IFN-I) responses for host defense against pathogens. Although T cells highly express STING, its physiological role remains unknown. Here, we show that costimulation of T cells with the STING ligand cGAMP and TCR leads to IFN-I production and strongly inhibits T-cell growth. TCR-mediated mTORC1 activation and sustained activation of IRF3 are required for cGAMP-induced IFN-I production, and the mTORC1 activity is partially counteracted by cGAMP, thereby blocking proliferation. This mTORC1 inhibition in response to costimulation depends on IRF3 and IRF7. Effector T cells produce much higher IFN-I levels than innate cells in response to cGAMP. Finally, we demonstrated that STING stimulation in T cells is effective in inducing antitumor responses in vivo. Our studies demonstrate that the outputs of STING and TCR signaling pathways are mutually regulated through mTORC1 to modulate T-cell functions.**

## Introduction

In addition to the antigen-specific TCR signals, T-cell activation is regulated by several different signals through costimulatory receptors. The most critical positive costimulatory signal is mediated by CD28 upon interaction with its ligands CD80/86 on APCs. By contrast, the inhibitory costimulatory receptors, cytotoxic T lymphocyte–associated protein 4 (CTLA-4) and programmed cell death 1 (PD-1) deliver negative signals to terminate T-cell responses and prevent autoimmune responses. The balance between these positive and negative costimulation signals determines the activation state, differentiation, and functions of T cells (Chen & Flies, 2013).

Mechanistic target of rapamycin (mTOR) is activated by TCR/CD28 signals and environmental signals and regulates cellular metabolism and protein synthesis through downstream pathways, such as 4E-BP1 and S6 kinase, and integrates these signals to regulate T-cell proliferation and differentiation (Chi, 2012). mTOR interacts with several proteins to form mTOR complex-1 (mTORC1) and -2 (mTORC2), which contain the essential scaffold protein Raptor and Rictor, respectively. T cell–specific gene deletion analyses revealed that mTORC1 has a central role for T-cell activation, differentiation, and antigen-specific immune responses in vivo (Yang et al, 2013).

Toll-like receptors (TLRs) are primary sensors in the innate immune system and recognize pathogen-associated molecular patterns (Takeda et al, 2003) to produce inflammatory cytokines and induce up-regulation of MHC and costimulatory molecules on APCs (Medzhitov, 2001). TLRs are also expressed by T cells, where they can have costimulatory functions. Indeed, TLR2 ligands enhance T-cell proliferation upon TCR stimulation (Komai-Koma et al, 2004; Cottalorda et al, 2006), directly trigger Th1 effector functions without TCR stimulation (Imanishi et al, 2007), and promote Th17 responses (Reynolds et al, 2010). Furthermore, we have shown that nucleic acids induce costimulation signals for Th2 differentiation independently of any known nucleic acid sensors, including TLRs, RIG-I–like receptors (RLRs), inflammasomes, and STING (Imanishi et al, 2014).

[1]Laboratory for Cell Signaling, RIKEN Center for Integrative Medical Sciences, Yokohama, Japan    [2]Department of Cell Signaling, Institute of Biomedical Sciences, Kansai Medical University, Hirakata, Japan    [3]Laboratory for Metabolomics, RIKEN Center for Integrative Medical Sciences, Yokohama, Japan    [4]Japan Agency for Medical Research and Development (AMED)-PRIME, Japan Agency for Medical Research and Development, Tokyo, Japan    [5]Graduate School of Medical Life Science, Yokohama City University, Yokohama, Japan    [6]Division of Molecular Genetics, Cancer Research Institute, Kanazawa University, Kanazawa, Japan    [7]Division of Innate Immunity, Department of Microbiology and Immunology, Institute of Medical Science, University of Tokyo, Tokyo, Japan    [8]Department of Cell Biology and the Sylvester Comprehensive Cancer Center, University of Miami Miller School of Medicine, Miami, FL, USA    [9]Division of Physiological Chemistry and Metabolism, Graduate School of Pharmaceutical Sciences, Keio University, Tokyo, Japan    [10]Laboratory of Vaccine Science, World Premier International Research Center Initiative (WPI) Immunology Frontier Research Center, Osaka University, Suita, Japan    [11]Laboratory of Adjuvant Innovation, National Institutes of Biomedical Innovation, Health and Nutrition, Osaka, Japan    [12]Laboratory of Host Defense, WPI Immunology Frontier Research Center, Osaka University, Osaka, Japan    [13]Laboratory for Cell Signaling, WPI Immunology Frontier Research Center, Osaka University, Osaka, Japan

Correspondence: takashi.saito@riken.jp; takayuki.imanishi@riken.jp
Wakana Kobayashi's present address is Laboratory for Mucosal Immunity, RIKEN Center for Integrative Medical Sciences, Yokohama, Japan.
Takayuki Hoshii's present address is Department of Pediatric Oncology, Dana-Farber Cancer Institute and Division of Hematology/Oncology, Boston Children's Hospital, Harvard Medical School, Boston, USA and Center for Epigenetics Research, Memorial Sloan Kettering Cancer Center, New York, USA

STING is a pattern recognition receptor localized in the ER membrane (Ishikawa & Barber, 2008) and recognizes cyclic dinucleotides (CDNs) derived from bacteria, resulting in induction of IFN-I responses (Burdette et al, 2011). STING also plays a central role in detecting cytosolic viral DNA (Ishikawa & Barber, 2008; Ishikawa et al, 2009). DNA derived from pathogens and even self-DNA (Gao et al, 2015) are recognized by the cyclic GMP-AMP (cGAMP) synthase (cGAS) (Sun et al, 2013), which catalyzes the conversion of GTP and ATP into the second messenger 2'3' cGAMP (Wu et al, 2013), which binds to and activates STING.

In this study, we assessed the function of STING in T cells and demonstrated that STING activation induces suppression of T-cell proliferation through inhibiting TCR-induced mTORC1 activation. STING-mediated inhibition of mTORC1 is dependent on IRF3/7 but not TBK1/IKK$\varepsilon$. We also found that naive T cells produce IFN-I upon STING and TCR stimulation. Mechanistically, TCR stimulation induces the sustained activation of IRF3 and provides the signals for mTORC1 activation for IFN-I responses. Our data show the central role of mTORC1 in STING-mediated proliferation inhibition and IFN-I responses in T cells. Finally, we demonstrated that STING in T cells is crucial for antitumor immune responses.

# Results

## Activation of STING in T cells inhibits growth

Naive CD4[+] T cells express STING protein at levels similar to BMDCs (Fig S1A), suggesting their intrinsic function in T cells as pattern recognition receptors. Whereas TLR ligands directly enhance T-cell proliferation upon TCR stimulation (Komai-Koma et al, 2004; Cottalorda et al, 2006), we found that STING ligands such as cGAMP and DMXAA strongly inhibit proliferation of naive CD4[+] T cells upon stimulation with anti-CD3/CD28 (Fig 1A). Studies with STING-deficient (KO) mice confirmed that this suppression is STING dependent. Similar results were obtained with naive CD8[+] T cells (Fig S1B). Notably, cGAMP inhibited T-cell proliferation without lipofection similarly to DMXAA, which has a cell-permeable structure. The inhibition of proliferation by cGAMP was also observed in an antigen-specific system, using T cells from Ovalbumin (OVA)-specific OT-II Tg mice (Fig S1C). Although proliferation was inhibited, the percentage of live cells in these cultures did not change in the presence of STING ligands except for those stimulated with high doses of DMXAA (Fig 1B). Consistently, only high concentrations of

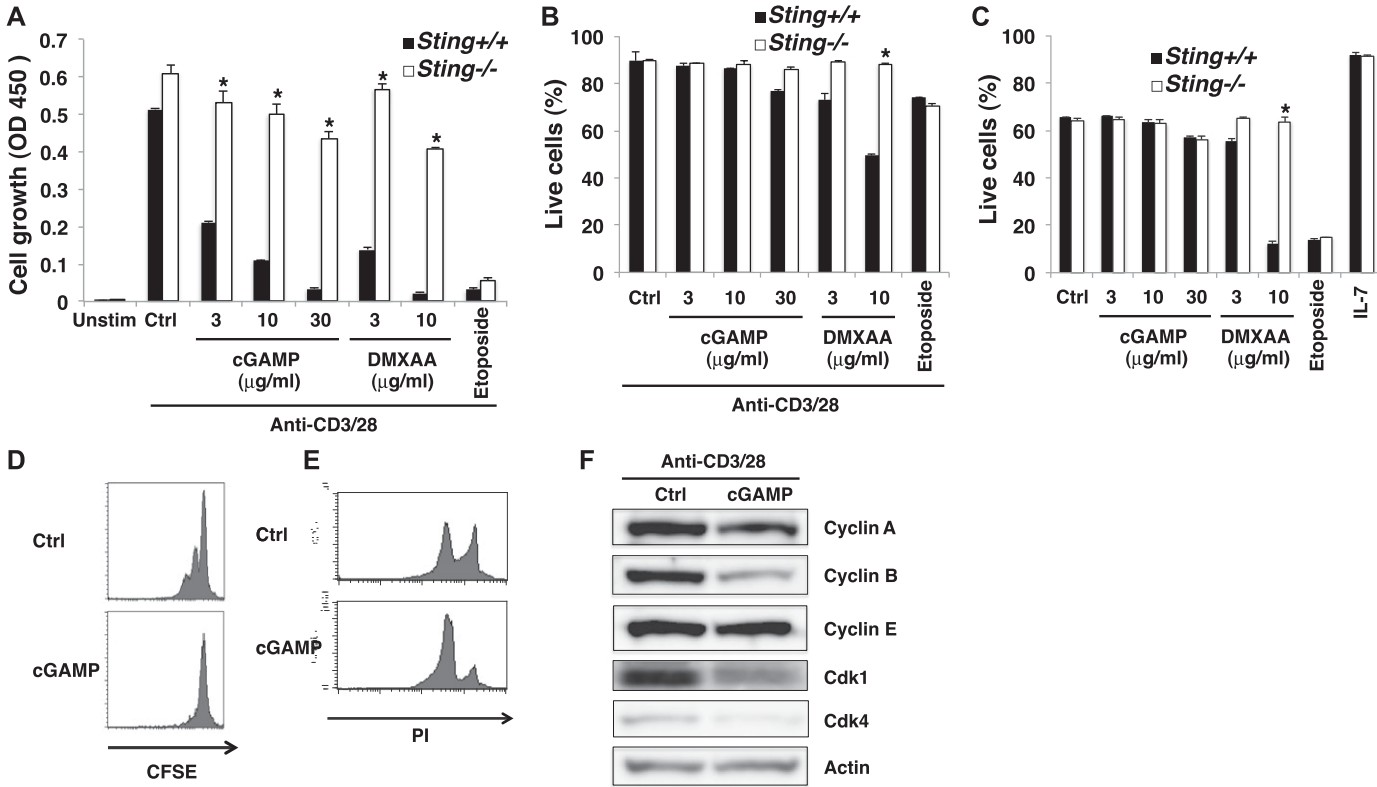

**Figure 1. STING stimulation in T cells inhibits cell cycle progression.**
**(A, B)** FACS-sorted naive CD4[+] T cells from *Sting*[+/+] or *Sting*[−/−] mice were stimulated with immobilized anti-CD3ε plus anti-CD28 (anti-CD3/CD28) Abs with or without STING ligands cGAMP or DMXAA. Cell growth was assessed after 48 h of stimulation by a WST-8 proliferation assay (A), and cell survival was determined after 18 h of stimulation by Propidium iodide (PI) and Annexin V staining (B). **(C)** Naive CD4[+] T cells from *Sting*[+/+] or *Sting*[−/−] mice were stimulated with cGAMP or DMXAA for 18 h, and cell survival was determined by PI and Annexin V staining. **(D, E)** Naive CD4[+] T cells were stimulated with anti-CD3/CD28 Abs with or without cGAMP for 48 h, and cell division was determined by a CFSE-labeling assay (D) and cell cycle was assessed by PI staining (E). **(F)** Western blot analysis for protein expression of cell cycle–related genes in CD4[+] T cells upon stimulation with anti-CD3/CD28 Abs with or without cGAMP for 24 h. Data are the mean from duplicate (A–C) ± SD. Data are representative of at least three independent experiments. **(A–F)** *P < 0.05, *t* test (compared with WT cells treated with indicated ligands).

DMXAA induced cell death as efficiently as etoposide in the absence of TCR stimulation (Fig 1C). We also found that treatment with a pan-caspase inhibitor Z-VAD or use of RIP3 (critical kinase responsible for necroptosis)-KO T cells did not affect the cGAMP-induced growth inhibition upon TCR stimulation (Fig S1D and E). These data suggest that cGAMP as the natural ligand for STING may inhibit proliferation of T cells through growth arrest rather than cell death. To test this possibility, naive CD4+ T cells were labeled with 5-(and -6)-Carboxyfluorescein diacetate succinimidyl ester (CFSE) and stimulated with anti-CD3/CD28 with or without cGAMP. Cell division was severely impaired by cGAMP (Fig 1D), and activated CD4+ T cells remained in the G0-G1 phase of the cell cycle in the presence of cGAMP (Fig 1E), indicating that costimulation of T cells with TCR and cGAMP induced cell cycle arrest. Consistently, mRNA and protein expression of cell cycle–related genes such as cyclins A2, B1, D2, Cdk1, and Cdk4 were reduced in the presence of cGAMP in STING-dependent fashion (Fig S1F and 1F). Conversely, the expression of CDK inhibitor p21 (Cdkn1a) and p27 Kip1 (Cdkn1b) was up-regulated

by cGAMP (Fig S1F and G). These data indicate that STING activation in T cells induces cell cycle arrest by modulating the expression of cell cycle–related genes.

## STING signals inhibit activation of mTORC1

Because mTORC1 signaling is required for cell cycle regulation in T cells through the induction of cell cycle–related genes (Yang et al, 2013), it seemed possible that the cell cycle arrest by cGAMP was mediated by the inhibition of mTORC1 activation upon TCR/CD28 stimulation. Indeed, cGAMP strongly inhibited the activation of mTORC1 downstream signaling molecules such as S6K1 and 4E-BP1 upon anti-CD3/28 stimulation (Fig 2A). By contrast, Akt activation was modestly enhanced (Fig 2A), which is similarly observed in Raptor-KO T cells (Yang et al, 2013). Interestingly, IL-2R signaling events, including phosphorylation of STAT5 and JAK3, were also inhibited (Fig 2A). We used STING-KO T cells to confirm that this cGAMP-induced inhibition of both mTORC1 and IL-2R signaling was

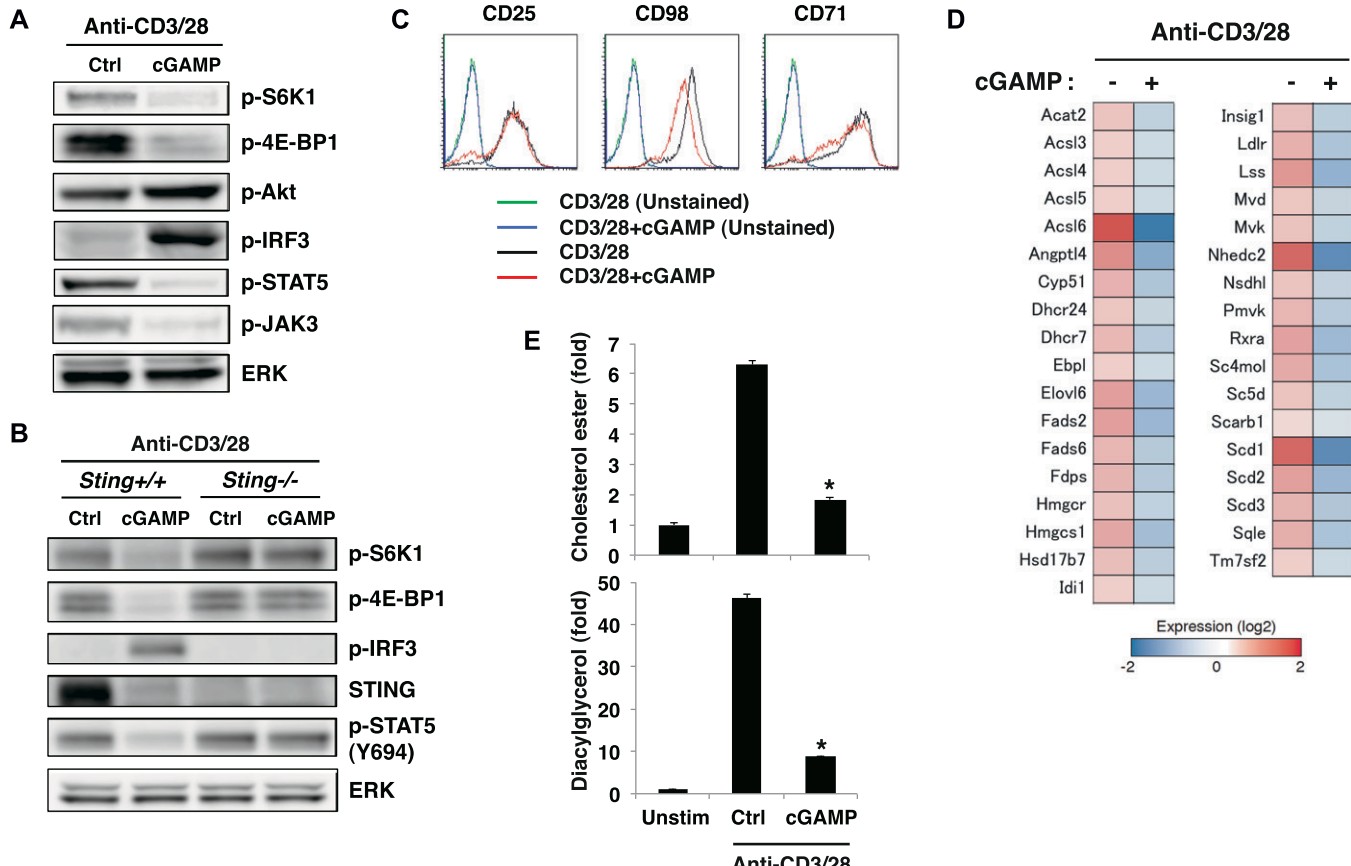

**Figure 2. STING activation leads to the inhibition of the mTORC1 pathway.**
**(A, B)** Western blot analysis of the indicated phosphorylated proteins in CD4+ T cells from *Sting*+/+ or *Sting*−/− mice upon stimulation with anti-CD3/CD28 Abs with or without cGAMP for 24 h. **(C)** FACS analysis of indicated surface molecules on CD4+ T cells upon stimulation with anti-CD3/CD28 Abs with or without cGAMP for 24 h. **(D)** RNA-seq data of the expression of lipid synthesis–related genes in CD4+ T cells upon stimulation with anti-CD3/CD28 Abs with or without cGAMP for 24 h. **(E)** Quantification of total cholesterol ester (upper) and total DAG (lower) in CD4+ T cells upon stimulation with anti-CD3/CD28 Abs with or without cGAMP for 24 h, assessed by LC-MS. Data are the mean from triplicate ± SD (E). Data are representative of at least three independent experiments (A–C). **(E)** *$P < 0.05$, t test (compared with that with anti-CD3/28 alone).

dependent on STING (Fig 2B). Although TCR-induced IL-2 production was slightly reduced by cGAMP upon TCR stimulation (Fig S2A), the addition of exogenous IL-2 failed to restore the inhibition of either cGAMP-induced cell growth or STAT5 activation (Fig S2B), suggesting that STING stimulation inhibits the activation of STAT5 independently of IL-2 production. Anti-IL-2R blocking Ab inhibited the phosphorylation of STAT5 but not 4E-BP1 (Fig S2C), indicating that STING activation by cGAMP inhibits TCR-induced mTORC1 signaling and IL-2-induced STAT5 activation. Notably, anti-IL-2R blocking Ab did not inhibit T-cell proliferation upon anti-CD3/28 stimulation in our experimental system, suggesting that cGAMP-induced inhibition of STAT5 activation was not required for STING-mediated growth inhibition (Fig S2D). We confirmed that other STING ligands, cyclic di-AMP and DMXAA, also inhibited both mTORC1 and IL-2R signaling (Fig S2E). mTORC1 regulates the expression of the amino acid transporter CD98 and the transferrin receptor CD71 (Yang et al, 2013), and we found that TCR-induced expression of both CD98 and CD71, but not CD25, was impaired by cGAMP (Fig 2C). These data indicate that cGAMP specifically inhibits both mTORC1 and IL-2 pathways upon TCR stimulation.

Pathway analysis of genes with down-regulation in CD4$^+$ T cells stimulated with anti-CD3/28 plus cGAMP as compared to those stimulated with anti-CD3/28 alone showed that cGAMP stimulation reduced the expression of lipid synthesis–related genes (Table 1 and Fig 2D), which are also regulated by mTORC1 (Yang et al, 2013). Of note, the reduced expression of those genes upon stimulation with TCR plus cGAMP was similarly observed in Raptor-KO T cells upon TCR stimulation (Fig S2F). Recently, it has been shown that mTORC1-induced cholesterol synthesis is critical for antigen-driven clonal expansion (Kidani et al, 2013). LC-mass spectrometry (LC-MS) analysis showed the total content of cholesterol esters was drastically reduced in T cells stimulated with anti-CD3/28 plus cGAMP as compared with T cells stimulated with anti-CD3/28 (Fig 2E). Lipin-1 (Han et al, 2006), the enzyme generating DAG, is a target of, and regulated by, mTORC1 (Eaton et al, 2013). MS analysis demonstrated that the total DAG was substantially reduced in T cells stimulated with TCR and cGAMP (Fig 2E). These data support the idea that the STING-mediated signal inhibits lipid synthesis through the inhibition of TCR-induced mTORC1 activation.

Altogether, STING activation induces the suppression of mTORC1 signaling and results in defective T-cell proliferation.

## CDNs induce type I IFN production from T cells in TCR stimulation–dependent manner

RNA-seq analysis revealed that the top 20 up-regulated genes in T cells upon stimulation with anti-CD3/28 plus cGAMP (Fig S3A) were all interferon-stimulated genes (ISGs) except for an unknown gene, AW011738, suggesting that T cells may induce IFN-I production by STING activation as innate cells. We found that cGAMP alone could induce the expression of IFN-I mRNA (Fig S3C) but failed to produce either IFN-$\beta$ (Fig 3A) or IFN-$\alpha$ (Fig S3B). However, when activated with anti-CD3/CD28, cGAMP and c-di-AMP induced IFN-I production from both naive CD4$^+$ and CD8$^+$ T cells (Figs 3A and S3B and S3D). Interestingly, robust production of type III IFN (IFN-$\lambda$2/3) was also observed (Fig S3E). IFN-I production correlated with the signal strength of TCR stimulation (Fig S3F). STING-induced IFN-I

**Table 1. Pathway-enrichment analysis.**

| | Count | P |
|---|---|---|
| Enrichment cluster 1 (score, 3.64) | | |
| Lipid biosynthesis | 14 | $5.1 \times 10^{-8}$ |
| Lipid metabolic process | 20 | $8.4 \times 10^{-6}$ |
| Lipid metabolism | 18 | $8.9 \times 10^{-6}$ |
| Fatty acid desaturase, type 1 | 5 | $9.4 \times 10^{-6}$ |
| Unsaturated fatty acid biosynthetic process | 5 | $3.3 \times 10^{-5}$ |
| Stearoyl-CoA 9-desaturase activity | 4 | $3.7 \times 10^{-5}$ |
| Fatty acid biosynthesis | 7 | $3.8 \times 10^{-5}$ |
| Fatty acid biosynthetic process | 8 | $4.7 \times 10^{-5}$ |
| Biosynthesis of unsaturated fatty acid | 6 | $5.1 \times 10^{-5}$ |
| Fatty acid metabolism | 9 | $1.0 \times 10^{-4}$ |
| Long-chain fatty acid biosynthetic process | 4 | $1.7 \times 10^{-4}$ |
| PPAR signaling pathway | 8 | $2.3 \times 10^{-4}$ |
| Palmitoyl-CoA 9-desaturase activity | 3 | $9.2 \times 10^{-4}$ |
| Monosaturated fatty acid biosynthetic process | 3 | $9.8 \times 10^{-4}$ |
| Enrichment cluster 2 (score, 3.42) | | |
| Nucleosome | 13 | $1.1 \times 10^{-8}$ |
| Negative regulation of cell proliferation | 14 | $1.5 \times 10^{-3}$ |
| Ubl conjugation | 31 | $2.8 \times 10^{-3}$ |
| Enrichment cluster 3 (score, 3.31) | | |
| ER | 34 | $1.2 \times 10^{-4}$ |
| Enrichment cluster 4 (score, 3.01) | | |
| Sterol biosynthetic process | 6 | $2.2 \times 10^{-5}$ |
| Cholesterol biosynthetic process | 6 | $5.2 \times 10^{-5}$ |
| Cholesterol biosynthesis | 5 | $7.5 \times 10^{-5}$ |
| Steroid biosynthesis | 6 | $7.5 \times 10^{-5}$ |
| Cholesterol metabolism | 6 | $4.5 \times 10^{-4}$ |
| Sterol metabolism | 6 | $7.8 \times 10^{-4}$ |
| Cholesterol metabolic process | 7 | $1.0 \times 10^{-3}$ |
| Steroid biosynthetic process | 6 | $1.2 \times 10^{-3}$ |
| Steroid metabolism | 6 | $1.7 \times 10^{-3}$ |
| Enrichment cluster 4 (score, 2.26) | | |
| Oxidoreductase | 19 | $5.3 \times 10^{-4}$ |
| Oxidoreductase activity | 19 | $6.0 \times 10^{-4}$ |
| Oxidation-reduction process | 19 | $3.0 \times 10^{-3}$ |
| NADP | 7 | $1.7 \times 10^{-3}$ |

Pathway-enrichment analysis of gene expression with down-regulation in CD4$^+$ T cells stimulated with anti-CD3/28 plus cGAMP for 24 h as compared with those stimulated with anti-CD3/28 alone. Data were analyzed with DAVID Bioinformatics Resources 6.8 and represent one experiment.

production in T cells was not induced within 24 h after stimulation (Fig 3A), although DCs can produce IFN-I within a few hours in response to STING ligands (Roth et al, 2014). Notably, DMXAA could not induce IFN-I production in T cells (Figs 3A and S3B).

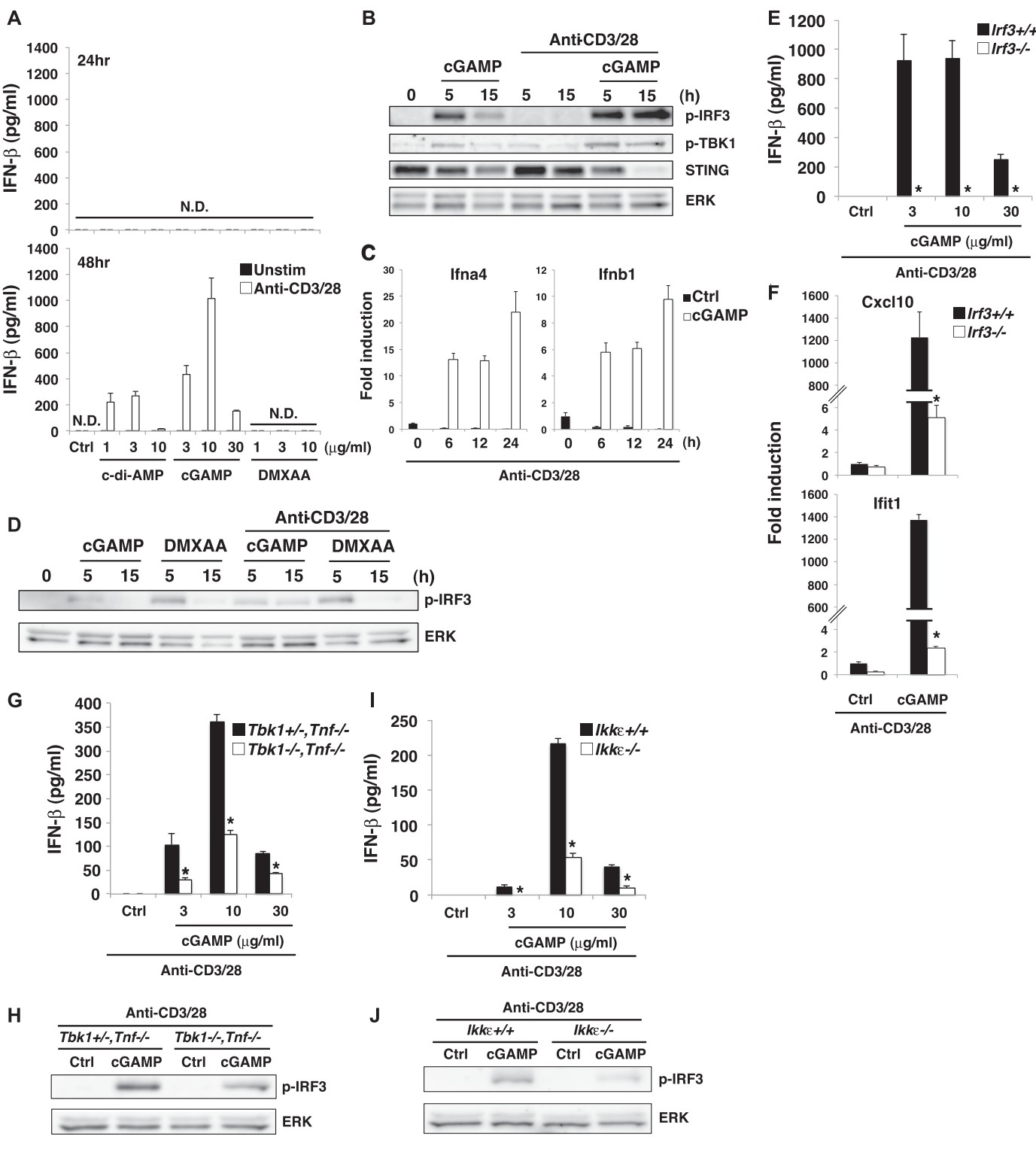

**Figure 3. TCR stimulation induces sustained activation of IRF3 upon CDN stimulation to produce type I IFNs.**
**(A)** Naive CD4⁺ T cells were stimulated with anti-CD3/CD28 Abs with or without the indicated STING ligands, and the level of IFN-β produced at 24 h (upper) and 48 h (lower) after stimulation was measured by ELISA. N.D., not detected < 2.0 pg/ml. **(B)** Western blot analysis of the activation status in CD4⁺ T cells upon stimulation with cGAMP in the presence or absence of stimulation with anti-CD3/CD28 Abs for 5 or 15 h. **(C)** qPCR analysis of the expression of IFN genes in CD4⁺ T cells upon stimulation with anti-CD3/CD28 Abs with or without cGAMP for the indicated time period. **(D)** Western blot analysis for IRF3 activation in CD4⁺ T cells upon stimulation with different STING ligands in the presence or absence of stimulation with anti-CD3/CD28 Abs. **(E)** Naive CD4⁺ T cells from *Irf3*⁺/⁺ or *Irf3*⁻/⁻ mice were stimulated with anti-CD3/CD28 Abs in the presence or absence of the indicated ligands, and IFN-β production was assessed by ELISA. **(F)** qPCR analysis of ISG genes in CD4⁺ T cells from *Irf3*⁺/⁺ or *Irf3*⁻/⁻ mice upon stimulation with anti-CD3/CD28 Abs with or without cGAMP for 24 h. **(G, H)** Naive CD4⁺ T cells from *Tbk1*⁺/⁻ *Tnf*⁻/⁻ or *Tbk1*⁻/⁻ *Tnf*⁻/⁻ mice were stimulated with anti-CD3/CD28 Abs,

To clarify the mechanism underlying cGAMP-induced IFN-I production upon TCR stimulation, we analyzed the activation of TBK1 and IRF3, which are essential for IFN-I induction in innate cells (Ishii et al, 2006; Takaoka et al, 2007). Whereas cGAMP stimulation alone induced transient phosphorylation of IRF3, cGAMP together with TCR stimulation induced sustained activation of IRF3 and TBK1 (Fig 3B). Consistently, sustained expression of IFN-I mRNA was observed in naive CD4$^+$ T cells stimulated with anti-CD3/28 plus cGAMP (Fig 3C). Although DMXAA activated IRF3 more strongly than cGAMP, the activation was just transient, even in the presence of TCR stimulation (Fig 3D). Together, these data suggest that sustained activation of IRF3 is required for IFN-I production by T cells. It is noted that sustained activation of IRF3 induced by cGAMP and TCR stimulation was observed as early as 15 h after stimulation (Fig 3B) when T-cell division was not yet induced, indicating that TCR-induced sustained phosphorylation of IRF3 is induced independently of cell division. To confirm the requirement of sustained IRF3 activation for the induction of IFN-I in T cells, IRF3-KO naive CD4$^+$ T cells were stimulated with anti-CD3/CD28 and cGAMP. The induction of IFN-I production and ISGs, such as CXCL10 and IFIT1, was completely eliminated in IRF3-KO CD4$^+$ T cells (Fig 3E and F). We next investigated the contribution of TBK1 to IFN-I responses by T cells. Unlike in innate cells, both IFN-I production and IRF3 activation were only partially impaired in TBK1-KO naive CD4$^+$ T cells (Fig 3G and H). Then, we analyzed the contribution of IKKε, which is a close homologue of TBK1 but only partially contributes to the induction of several ISGs such as CCL5 and CCL2 in innate cells (Ishii et al, 2006). We found that T cells highly express IKKε (Fig S3G) and that both IFN-I production and IRF3 activation were partially reduced in IKKε-KO CD4$^+$ T cells (Fig 3I and J). These data suggest that, unlike innate cells, both TBK1 and IKKε are equally and redundantly important for IFN-I production by T cells through the activation of IRF3.

**Effector T cells produce robust type I IFNs**

We previously reported that TLR2 ligands directly induce IFN-γ production by effector Th1 and CD8$^+$ T cells even without TCR stimulation and that this is enhanced by IL-2 (Imanishi et al, 2007). We assumed a similar possibility that STING ligands might also directly stimulate IFN-I production from effector T cells in the absence of TCR stimulation. As expected, cGAMP induced IFN-I production from Th1 cells and activated CD8$^+$ T cells even in the absence of TCR stimulation, and this was further enhanced by IL-2 and strongly augmented by TCR stimulation (Figs 4A and S4A). Importantly, the amount of IFN-I produced by activated CD8$^+$ T cells upon stimulation with TCR and cGAMP (12 ng/ml, Fig.4A) was 10-fold higher than that from BMDCs (1–2 ng/ml, Fig S4B) (Figs 4A and B, and S4A and B). Notably, unlike naive T cells, effector T cells can produce IFN-I within 24 h in response to cGAMP. To elucidate the molecular mechanisms underlying direct induction of IFN-I production from effector T cells by cGAMP, we analyzed the signaling molecules downstream of STING in effector

T cells. Whereas cGAMP alone transiently induced the activation of IRF3 in naive T cells (Fig 3B), sustained activation of IRF3 was induced in effector Th1 cells and activated CD8$^+$ T cells (Fig 4C and D). The activation of IRF3 by cGAMP was augmented by IL-2 or TCR stimulation (Fig 4C and D) and correlated with the IFN-I production. These data indicate that cGAMP alone stimulates IFN-I production from effector/activated T cells by inducing the sustained activation of IRF3, which is further enhanced by IL-2 or TCR stimulation. We also analyzed the other adaptive immune cells, B cells, and found that LPS or anti-IgM plus cGAMP stimulation did not induce IFN-I production by B cells, but cGAMP inhibited LPS or anti-IgM induced B-cell proliferation similarly to TCR-activated T cells (Fig S4C). Unlike in T cells, cGAMP transiently induced expression of IFN-I mRNA upon LPS plus cGAMP stimulation of B cells (Fig S4D).

**IRF3/7 are required for the STING-mediated inhibition of mTORC1 and IL-2 signaling**

Because T cells can produce IFN-I in response to cGAMP and TCR stimulation, it is possible that IFN-I produced by the T cells may inhibit the activation of mTORC1 and the proliferation. This issue was addressed by analyzing these signals in IFNα receptor 1 (IFNAR1)-KO CD4$^+$ T cells. Phosphorylation of S6K1, 4E-BP1, and STAT5 was similarly inhibited by cGAMP, whereas cGAMP-induced growth inhibition was partially restored in IFNAR1-KO T cells (Fig 5A and B), indicating that IFN-I signaling is partly involved in growth inhibition but not in inhibition of mTORC1 and IL-2 signaling pathways. We confirmed that cGAMP-induced growth inhibition was also partly blocked in the presence of rapamycin, an mTOR inhibitor (Fig S5A), indicating that both the inhibition of mTORC1 activation and IFN-I signaling are involved in STING-mediated growth inhibition. In addition, it is noted that the simultaneous treatment with both rapamycin and anti-IFNAR1 blocking Ab did not completely block the cGAMP-induced growth inhibition (Fig S5A), suggesting that yet unidentified pathway is involved in STING-mediated growth inhibition.

Next, we sought to determine the mechanism of STING-mediated inhibition of both mTORC1 and IL-2 pathways. We first examined the involvement of TBK1 and IKKε in the cGAMP-induced inhibition of mTORC1 and IL-2 pathways. Surprisingly, cGAMP-induced growth inhibition was equivalent in CD4$^+$ T cells from TBK1- and IKKε-KO mice (Fig 5C). We then analyzed the involvement of IRF3/7 in this process. cGAMP-induced growth inhibition was partially restored in IRF3-KO CD4$^+$ T cells (Fig 5C) and in IRF7-KO CD4$^+$ T cells (Fig S5B) and more strongly restored in IRF3/7-doubly deficient (DKO) CD4$^+$ T cells (Fig 5D). In addition, partial restoration of cGAMP-induced inhibition of mTORC1 (p-S6K1 and p-4E-BP1) and IL-2 (p-STAT5) signaling was observed in IRF3/7-DKO CD4$^+$ T cells (Figs 5E and S5C). Consistently, the inhibition of the expression of lipid synthesis–related genes, CD98 and CD71, by cGAMP was partly recovered in IRF3/7-DKO CD4$^+$ T cells (Figs 5F and S5D). Particularly, the partial restoration of cGAMP-induced growth inhibition in IRF3-KO CD4$^+$ T cells was

with or without cGAMP, and IFN-β production was assessed by ELISA (G), and phosphorylation of IRF3 was assessed by Western blot analysis (H). **(I, J)** Naive CD4$^+$ T cells from *Ikkε*$^{+/+}$ or *Ikkε*$^{-/-}$ mice were stimulated with anti-CD3/CD28 Abs with or without cGAMP, and IFN-β production was assessed by ELISA (I), and phosphorylation of IRF3 was assessed by Western blot analysis **(J)**. Data are the mean from duplicate (A, E, G, I) or triplicate (C, F) ± SD. Data are representative of at least three independent experiments. **(A—J)** *$P < 0.05$, $t$ test (compared with WT cells treated with cGAMP).

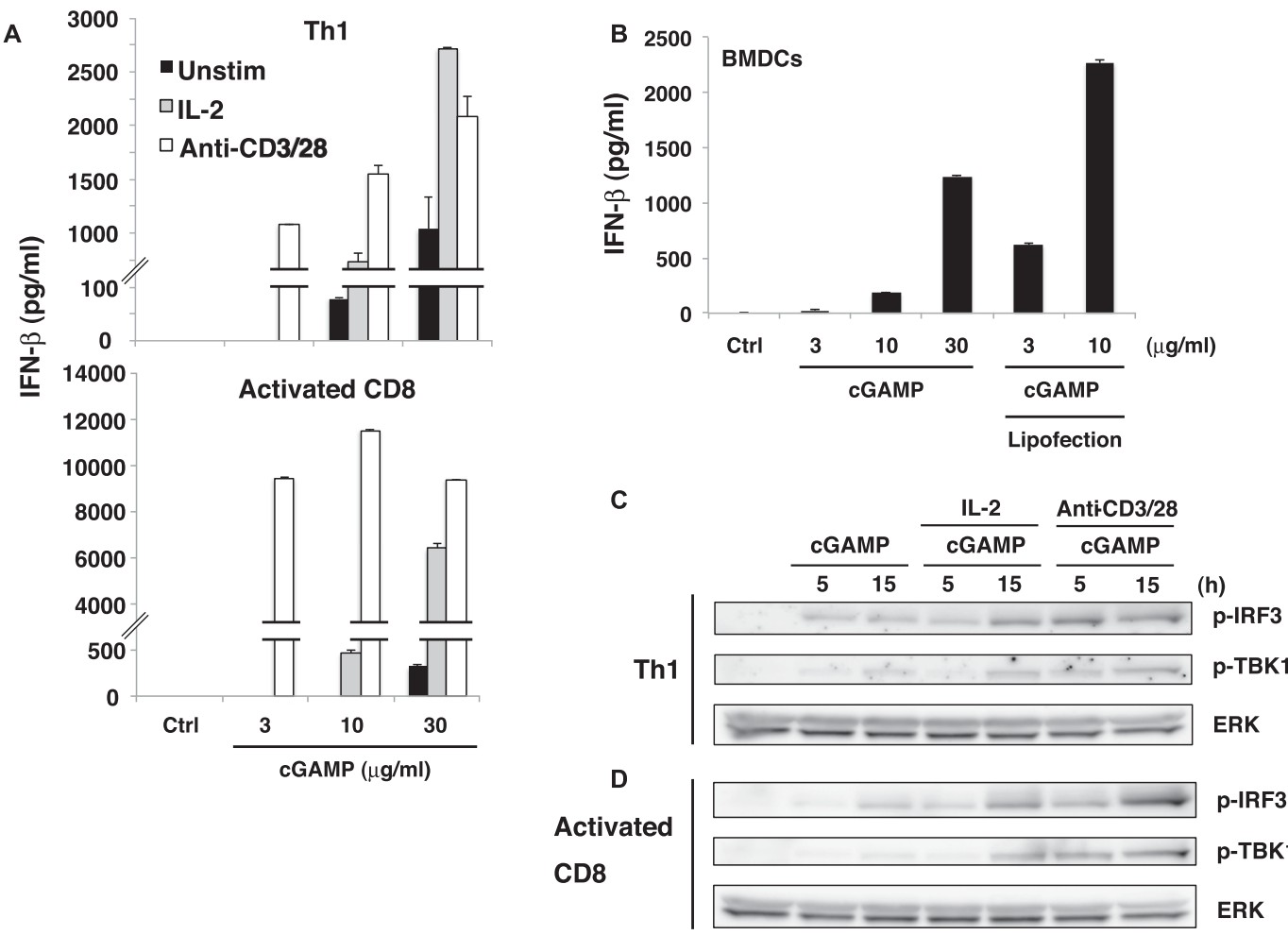

**Figure 4. Greatly enhanced STING-mediated type I IFN responses in effector/activated T cells.**
**(A)** Th1 cells or activated CD8⁺ T cells were stimulated with cGAMP in the presence (IL-2) or absence (unstim) of IL-2 or stimulation with anti-CD3/CD28 Abs for 24 h, and IFN-β production was assessed by ELISA. **(B)** cGAMP was added to the media or introduced by lipofection into BMDCs for 24 h, and IFN-β production was assessed by ELISA. **(C, D)** Western blot analysis of phosphorylation of IRF3 and TBK1 in Th1 cells (C) or activated CD8⁺ T cells (D) upon stimulation with cGAMP with or without IL-2 or anti-CD3/CD28 stimulation. Data are the mean from duplicate ± SD (A, B). Data are representative of at least three independent experiments (A—D).

completely cancelled by the addition of exogenous IFN-β (Fig S5E), whereas the addition of exogenous IFN-β partially cancelled it in IRF3/7-DKO CD4⁺ T cells (Fig S5F). In addition, cGAMP-induced inhibition of mTORC1 activation was largely intact in IRF3-KO CD4⁺ T cells (Fig S5G), suggesting that IRF3 and IRF7 have redundant function in STING-mediated T-cell growth inhibition. These data also suggest that normal cGAMP-induced growth inhibition in TBK1- or IKKε-KO T cells may be due to the remaining IFN-I production because cGAMP-induced IFN-I production was completely diminished in IRF3-KO T cells (Fig 3E) but only partially impaired in TBK1- or IKKε-KO T cells (Fig 3G and I).

We also analyzed whether cGAMP treatment may affect ER stress pathways because it has been shown that STING departure from ER causes ER stress that inhibits mTOR pathway (Moretti et al, 2017). However, phosphorylation of PERK, IRE1α, and eIF2α, which represent ER stress transducers, was not altered by cGAMP treatment (Fig S5H), indicating that inhibition of mTOR was not induced by ER stress but by STING signals in T cells.

Collectively, these data indicate that the TBK1/IKKε-IRF3/7-IFN-I axis inhibits T-cell proliferation and IRF3/7 are also critical for STING-mediated inhibition of mTORC1 and IL-2 pathways independently of TBK1/IKKε.

### TCR-induced activation of mTORC1 is required for STING-mediated type I IFN production

Although our data verified that the STING-IRF3/7 axis inhibits T-cell proliferation through blocking mTORC1 function, the involvement of mTORC1 in STING-mediated IFN-I production remains unclear. When naive CD4⁺ T cells were stimulated with anti-CD3/CD28 plus cGAMP in the presence of rapamycin, surprisingly, cGAMP-induced IFN-I production and the expression of ISGs were completely abrogated (Figs S6A and 6B). To confirm the importance of mTORC1 in STING-mediated IFN-I responses, we analyzed Raptor-KO CD4⁺ T cells because Raptor is an essential component of the mTORC1 complex. Similar to rapamycin-treatment, cGAMP-induced IFN-I

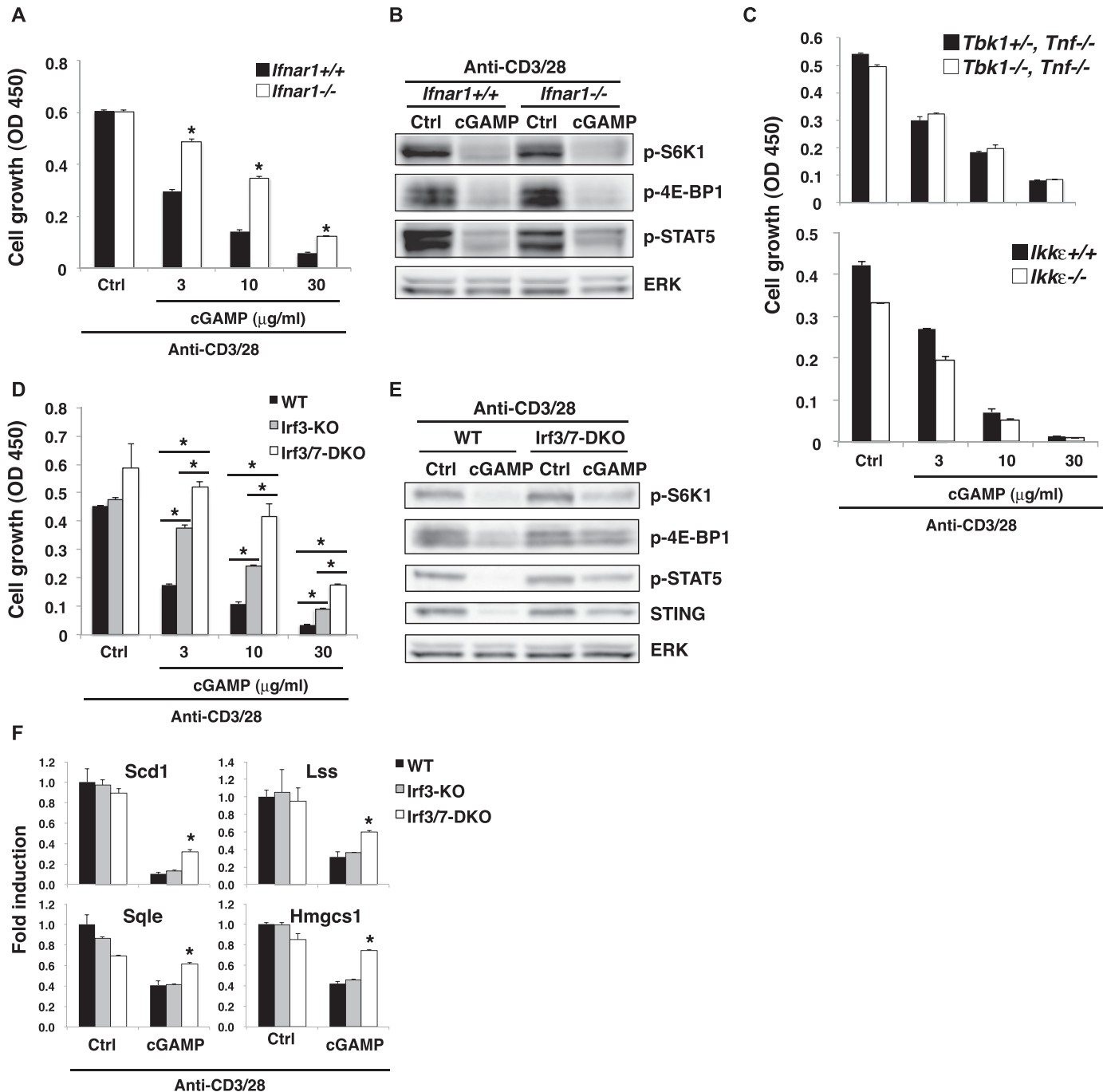

**Figure 5. Molecular mechanism of STING-mediated inhibition of the mTORC1 pathway.**
**(A, D)** Naive CD4[+] T cells from *Ifnar1*[+/+] or *Ifnar1*[−/−] mice (A) or WT, *Irf3*[−/−], or *Irf3*[−/−] *Irf7*[−/−] mice (D) were stimulated with anti-CD3/CD28 Abs with or without cGAMP for 48 h, and cell growth was assessed by a WST-8 cell proliferation assay. **(B, E)** Activation of the mTORC1 pathway was analyzed by Western blot in CD4[+] T cells from *Ifnar1*[+/+] or *Ifnar1*[−/−] mice (B) or *Irf3*[−/−] *Irf7*[−/−] (DKO) mice (E) upon stimulation with anti-CD3/CD28 Abs with or without cGAMP for 24 h. **(C)** Naive CD4[+] T cells from *Tbk1*[+/−] *Tnf*[−/−] or *Tbk1*[−/−] *Tnf*[−/−] mice (upper) or *Ikkε*[+/+] or *Ikkε*[−/−] mice (lower) were stimulated with anti-CD3/CD28 Abs with or without cGAMP, and cell growth was assessed by a WST-8 cell proliferation assay. **(F)** qPCR analysis of various genes in CD4[+] T cells from WT (closed bar), *Irf3*[−/−] (grey bar), or *Irf3*[−/−] *Irf7*[−/−] (open bar) mice upon stimulation with anti-CD3/CD28 Abs with or without cGAMP for 24 h. Data are the mean from duplicate ± SD (A, C, D) or triplicate (F) ± SD. Data are representative of at least three independent experiments (C—F). **(A, F)** *P < 0.05, t test (compared with WT cells treated with cGAMP). **(D)** *P < 0.05, t test (compared with WT cells or Irf3-KO cells treated with cGAMP).

production was completely abrogated in Raptor-KO CD4[+] T cells (Fig 6A). Quantitative PCR (qPCR) analysis revealed that the expression of type I and type III IFNs and ISGs, except for Ccl5, was severely impaired in Raptor-KO T cells (Fig 6B) and by rapamycin treatment

(Fig S6B), suggesting that mTORC1 signaling specifically regulates the induction of STING-mediated genes. Because TCR stimulation strongly activates the mTORC1 pathway and induces the sustained phosphorylation of IRF3, which is essential for the induction of IFN-I

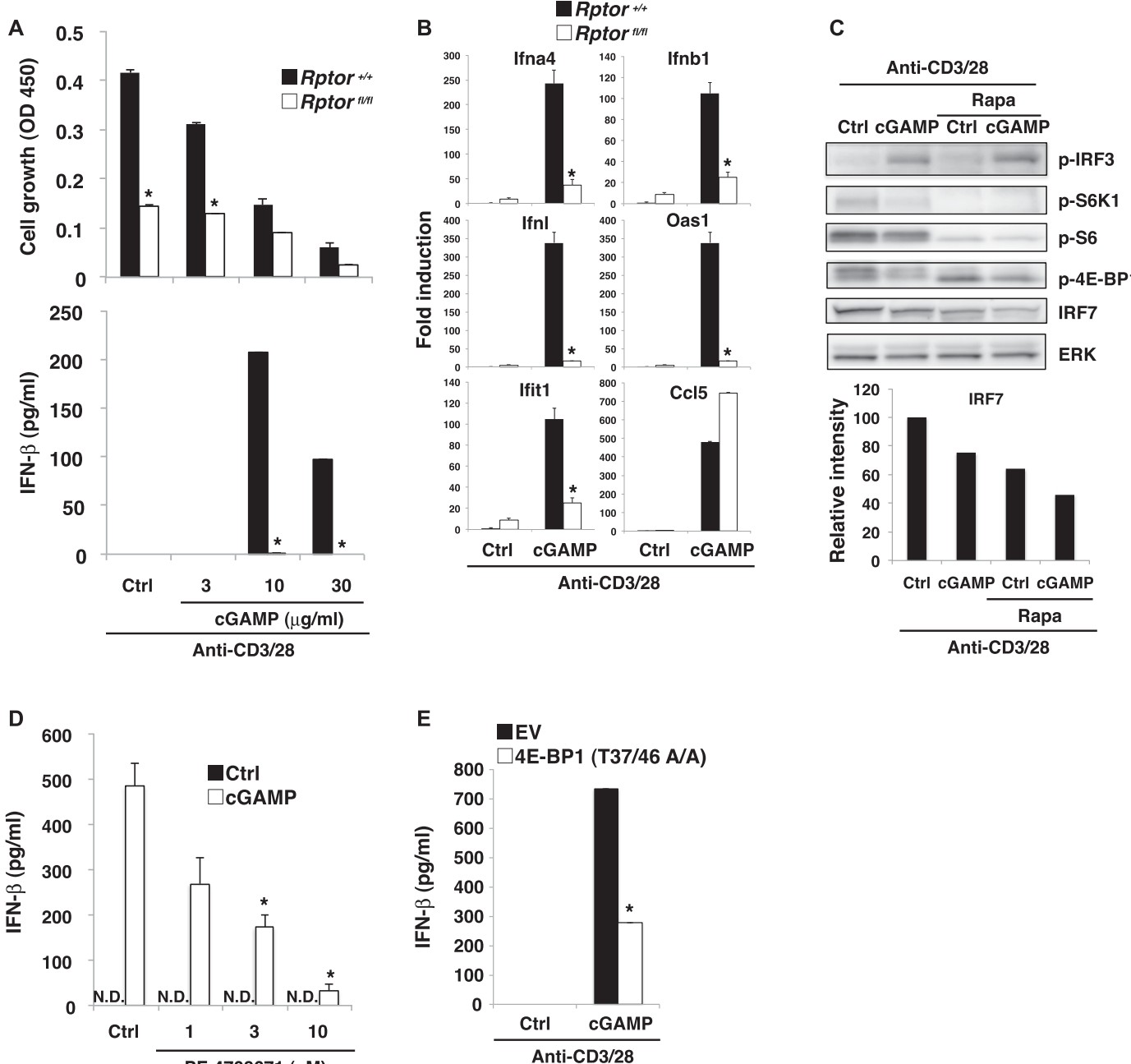

**Figure 6. mTORC1 signaling is required for STING-mediated type I IFN responses.**
**(A)** Naive CD4⁺ T cells from *Rptor⁺/⁺* (closed bar) or *Rptor^fl/fl* (open bar) mice were stimulated with anti-CD3/CD28 Abs with or without cGAMP for 48 h, and cell growth (upper) and IFN-β production (lower) were assessed by a WST-8 cell proliferation assay and ELISA, respectively. **(B)** qPCR analysis of ISG genes in CD4⁺ T cells from *Rptor⁺/⁺* (closed bar) or *Rptor^fl/fl* (open bar) mice upon stimulation with anti-CD3/CD28 Abs with or without cGAMP for 24 h. **(C)** Western blot analysis of activation of IRF3 and mTORC1-related molecules in CD4⁺ T cells with or without rapamycin pretreatment and upon stimulation with anti-CD3/CD28 Abs with or without cGAMP for 24 h (upper). Relative intensity of IRF7 was shown (lower). **(D)** Naive CD4⁺ T cells were untreated or pretreated with S6K inhibitor PF-4708671 and stimulated with anti-CD3/CD28 Abs with or without cGAMP for 48 h, and IFN-β production was assessed by ELISA. N.D., not detected < 2.0 pg/ml. **(E)** CD4⁺ T cells were retrovirally transduced with an empty vector (EV) or a 4E-BP1 (T37/46 A/A) construct and stimulated with anti-CD3/CD28 Abs with or without cGAMP for 24 h, and IFN-β production was assessed by ELISA. Data are the mean from duplicate (A, D, E) or triplicate (B) ± SD. Data are representative of at least three independent experiments (C–E). **(A, B)** *P < 0.05, t test (compared with WT cells treated with cGAMP). **(D)** *P < 0.05, t test (compared with that with anti-CD3/28 alone). **(E)** *P < 0.05, t test (compared with control cells treated with cGAMP).

and ISGs by cGAMP (Figs 3B, 3E, and F), we assumed that TCR-induced mTORC1 activation may be critical for the sustained phosphorylation of IRF3. However, rapamycin did not alter IRF3 activation by cGAMP plus TCR stimulation in CD4⁺ T cells (Fig 6C), indicating that TCR-induced mTORC1 signaling is critical for STING-mediated induction of IFN-I and ISGs independently of IRF3

activation. It is noted that the inhibition of T-cell growth by rapamycin treatment was similarly induced to the simultaneous treatment with both rapamycin and anti-IFNAR1 blocking Ab (Fig S5A) because rapamycin treatment inhibited both mTORC1 activation and cGAMP-induced IFN-I production.

In that case, how does mTORC1 signaling affect STING-mediated IFN-I responses? It has been reported that 4E-BP1/2 negatively regulates IFN-I production via translational repression of IRF7 mRNA in innate cells (Colina et al, 2008). In fact, IRF7 protein expression was slightly reduced in CD4+ T cells treated with rapamycin (Fig 6C), suggesting that mTORC1 is critical for IFN-I production by T cells stimulated with anti-CD3/28 plus cGAMP partly at least through translational control of IRF7. Then, we analyzed the involvement of S6K1 and 4E-BP1 as the main targets of mTORC1 in STING-mediated IFN-I production. PF-4708671, a specific inhibitor of S6K1 (Pearce et al, 2010), inhibited cGAMP-induced IFN-I production by CD4+ T cells upon TCR stimulation in a dose-dependent manner (Fig 6D). We also found that T cells overexpressing the dominant-negative 4E-BP1 (Thr-37/46 to alanine) produced less IFN-I (Fig 6E), indicating that both axes of mTORC1-S6K1 and mTORC1-4E-BP1 are critically involved in STING-mediated IFN-I responses in T cells upon TCR stimulation.

Although cGAMP inhibits mTORC1 activation, which is required for IFN-I production by T cells, cGAMP induces but does not inhibit IFN-I production. This paradox appears to be regulated by a quantitative balance between STING and mTORC1 signaling. Increasing doses of cGAMP induce stronger activation of IRF3 and higher production of IFN-I (Fig S6D and E). By contrast, the activation of the mTORC1 pathway, 4E-BP1 and S6K1, was inhibited by cGAMP in a dose-dependent manner (Fig S6E). 4E-BP1 was strongly inhibited, whereas S6K1 was weakly inhibited by cGAMP as compared with rapamycin (Fig S6C), suggesting that the S6K1 pathway may contribute to the induction of IFN-I production. Consistently, rapamycin completely inhibited IFN-I production through complete inhibition of both 4E-BP1 and S6K1 phosphorylation, whereas cGAMP mediated relatively weak inhibition of S6K1 (Fig S6F). In addition, the week inhibition of S6K1 activation by treatment with 0.1 nM rapamycin partially inhibited cGAMP-induced IFN-I production, whereas the same treatment inhibited T-cell proliferation as strongly as higher concentrations of rapamycin (Fig S6G). Therefore, it is likely that partial inhibition of S6K1 pathway by cGAMP allows the cGAMP-induced IFN-I production through the remaining activity of S6K1 and the downstream molecules of S6K1. Together, these findings indicate that activation of STING induces the partial inhibition of

mTORC1 activation, which is sufficient to inhibit T-cell proliferation on the one hand, whereas this partially remained activation of mTORC1 signals is critical for IFN-I production on the other hand (Fig S7).

### Critical role for STING in T cells in antitumor responses

It has been recently demonstrated that activation of the STING pathway is critical for antitumor immune responses in vivo (Woo et al, 2014; Deng et al, 2014). Indeed, STING-KO mice show impaired antitumor responses to radiation and immune checkpoint blockade therapies, such as PD-1/PD-L1 and CTLA4 (Woo et al, 2014; Deng et al, 2014; Demaria et al, 2015; Wang et al, 2017). Administration of STING ligands inhibits tumor growth and potentiates the antitumor effects of radiation and immune checkpoint blockade through the production of IFN-I (Deng et al, 2014; Demaria et al, 2015; Temizoz et al, 2015). Our finding that activated CD8+ T cells produce much higher levels of IFN-I than innate cells such as BMDCs (Fig 4A) raises the possibility that STING expressed in T cells may contribute to antitumor immune responses. To test this possibility, we made T cell–specific STING-KO mice by reconstituting RAG1-KO mice with STING-KO T cells and WT B cells. The mice were then inoculated with B16 melanoma cells, followed by injection of cGAMP intratumorally on day 8, 10, and 13 after tumor inoculation, and the tumor growth and survival were monitored. Tumor growth was accelerated in T cell–specific STING-KO ($Sting^{-/-}$) mice as compared with control mice ($Sting^{+/-}$), which had received both $Sting^{+/-}$ T cells and WT B cells (Fig 7A). Consistently, the survival of the tumor-bearing T cell–specific STING-KO mice were significantly lower than the control mice (Fig 7B).

These results suggest that STING expressed in T cells plays a crucial role in antitumor immunity.

## Discussion

Our studies have demonstrated that STING activation in T cells induces the suppression of T-cell proliferation through the inhibition of the mTORC1 pathway and the IFN-I signaling. This is the first report showing a functional link between the STING pathway and mTOR, the metabolic checkpoint kinase. STING-mediated inhibition of the mTORC1 pathway may be beneficial for host defense because the inhibition of pathogen-infected T-cell

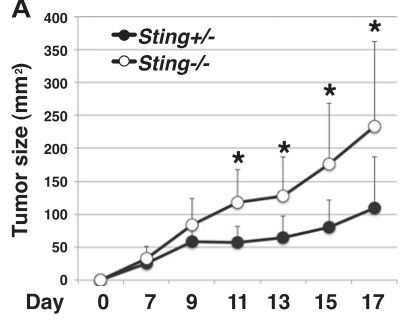
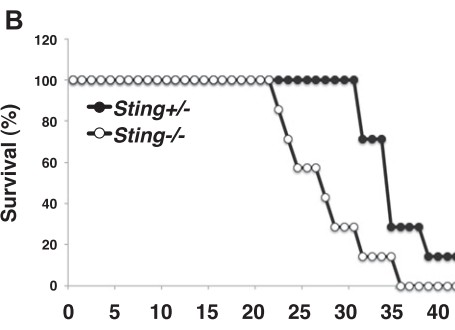

**Figure 7. T cell–intrinsic STING is critical for antitumor immunity.**
**(A, B)** T cells from $Sting^{+/-}$ or $Sting^{-/-}$ mice and B220+ cells from WT mice were cotransferred into Rag1-KO mice ($Sting^{+/-}$, $Sting^{-/-}$), and the mice were subcutaneously inoculated with $2 \times 10^5$ B16 F10 melanoma cells (n = 7 mice per group). On days 8, 10, and 13, the mice were subjected to intratumor injection of cGAMP and then monitored for tumor growth for 17 d (A) and for survival for 40 d (B). **(A)** *$P < 0.05$, $t$ test.

growth could block pathogen replication. Viruses have evolved to modify various cell signaling pathways in the host to establish optimal environments for their replication and spread. Most viruses induce glycolysis, fatty acid synthesis, and glutaminolysis in host cells to increase the energy supply for their replication (Sanchez & Lagunoff, 2015). mTORC1 is critical for TCR-induced glycolysis and fatty acid synthesis, and glutaminolysis activates the mTORC1 pathway (Duran et al, 2012). Therefore, STING-mediated inhibition of mTORC1 might be detrimental to virus replication and thus an adaptive host strategy to suppress virus replication.

Our data show that STING-mediated inhibition of mTORC1 partly requires IRF3/7 but not TBK1/IKKε in T cells. Because TBK1/IKKε-mediated phosphorylation of IRF3/7 is required for the induction of IFN-I genes in T cells, it is possible that IRF3/7 may inhibit the activation of mTORC1 independently of their function as transcription factors. Such function of IRF3 has been reported that IRF3 induces viral apoptosis through the interaction of its BH3-like domain with the pro-apoptotic protein Bax (Chattopadhyay et al, 2010). In addition, it has been reported that IRF3-mediated apoptosis requires linear polyubiquitination of IRF3 by LUBAC through TRAFs but not TBK1 (Chattopadhyay et al, 2016). Therefore, it is likely that IRF3/7 have a unique function other than transcription factors to suppress the mTORC1 activation in T cells because TBK1, a transcriptional regulator for IRF3, is not involved in STING-mediated inhibition of mTORC1 pathway. The mechanism of IRF3/7 to inhibit mTORC1 remains to be further investigated. Because STING-mediated mTORC1 inhibition was not completely restored in IRF3/7-DKO T cells, other molecules including other members of the IRF family, for example, IRF1, 5 and 9, may have functional redundancy with IRF3/7.

The most surprising finding in this study was that T cells produce IFN-I upon stimulation with STING and TCR, because T and B cells have been believed not to produce IFN-I. STING ligands induce robust IFN-I production by T cells similarly to innate cells. A most critical observation was that IFN-I production requires TCR stimulation. Our data clearly demonstrated that the activation of both IRF3 and mTORC1 is essential for STING-mediated IFN-I production. Because TCR stimulation triggers two events in this pathway, induction of the sustained activation of IRF3, and the activation of the mTORC1 pathway, TCR signaling is required to induce IFN-I responses upon STING activation. The reason why STING-mediated IFN-I production from naive T cells is induced later than innate immune cells is because mTOR activation is required for IFN-I production in T cells. mTORC1 activation is induced at the peaks later than 24 h after TCR stimulation. Consistently, activated effector T cells, in which mTORC1 pathway is already activated, produce type IFN-I shortly within 24 h in response to cGAMP. Because neither TCR stimulation nor STING activation alone induces IFN-I production, this regulation system allows only antigen-specific activated T cells to produce IFN-I. IFN-I has been shown to exhibit blocking functions in infectious and autoimmune diseases and cancer development. Therefore, IFN-I specifically produced by antigen-specific T cells may play roles in preventing the development of these diseases at inflammatory sites.

We defined here for the first time the reciprocal regulation between STING and the mTORC1 pathway for the modulation of T-cell functions, particularly induction of growth arrest and IFN-I production. STING-mediated mTORC1 activation together with TCR stimulation induced growth inhibition on the one hand and IFN-I production on the other hand. Because complete blockade of mTOR function by rapamycin completely inhibits IFN-I production, STING ligand–induced inhibition of mTORC1 is not complete with the remaining signals through S6K1, which we found contributes the induction of IFN-I production.

We have demonstrated that STING-mediated responses in T cells vary depending on species and doses of different STING agonists. Indeed, cGAMP and c-di-AMP as CDNs, but not DMXAA, induced IFN-I production upon T-cell stimulation. Conversely, DMXAA but not cGAMP and c-di-AMP induces T-cell death. Consistently, a recent study reported that activation of STING in T cells by DMXAA induces cell death (Larkin et al, 2017). Moreover, even among CDNs, we found that cGAMP induces more IFN-I production than c-di-AMP. Recent studies demonstrated that natural variant alleles of STING, namely, the R232H variant of human STING and the R231A variant of mouse STING, were activated by cGAMP but not c-di-GMP (Diner et al, 2013). Furthermore, it has been reported that DMXAA, which specifically binds to mouse STING, also activates human STING bearing a unique point mutation (S162A) at the CDN-binding site (Gao et al, 2013). These findings suggest that different STING agonists bind to different sites of STING to activate downstream signaling. Therefore, the effect of DMXAA on T cells may be quite different from that of cGAMP and c-di-AMP. Recently, it has been reported that the cell-permeable small molecule STING agonist 10-carboxymethyl-9-acridanone (CMA) induces T-cell apoptosis, whereas CMA does not induce IFN-I production (Gulen et al, 2017), suggesting the possibility that the binding site of CMA to STING may be similar to DMXAA. Indeed, CMA and DMXAA are structurally similar, and CMA activates IFN-I responses in murine cells but not human cells such as DMXAA (Cavlar et al, 2013). It has been reported that the duration and the magnitude of FcRγ signals determine mast cell survival and degranulation, respectively (Yamasaki et al, 2004). Prolonged ERK activation induced by antigen alone induces survival but not degranulation. By contrast, transient strong ERK activation induced by IgE plus antigen induces degranulation but not survival. Therefore, it is possible that different STING agonists induce different activation status of STING, leading to different outcomes. Considering the different characteristics of the ligands, the development of new STING agonists and antagonists with the best activity as vaccines or IFN-I inducers for immunotherapy of cancer and autoimmune disorders could be achieved.

cGAMP has been shown to provide strong antitumor effects in immune-competent mice. It has been thought that cGAMP-enhanced cross-presentation of tumor-associated antigens in DCs to CD8[+] cytotoxic T cells is one of the mechanisms underlying STING ligand–induced antitumor immunity (Wang et al, 2017). STING is required for radiation-induced antitumor T-cell responses, which are dependent on IFN-I signaling in DCs (Deng et al, 2014). Furthermore, immune-checkpoint therapy using anti-PD-1Ab is effective in the presence of STING activation (Woo et al, 2014; Demaria et al, 2015; Wang et al, 2017). In the present study, we showed that T cell–intrinsic STING is important for the induction of cGAMP-induced antitumor effects. It has been reported that IRF7/IFN-β activation enhances chimeric antigen receptors (CARs) T cell–mediated

antitumor activity (Zhao et al, 2015), suggesting that in addition to IFN-I from innate cells, as is widely believed, STING-mediated production of IFN-I by T cells might contribute significantly to antitumor immunity. It is worth considering that antigen-specific effector T cells are localized in the tumor microenvironment, where the effector cells receive STING activation signals and produce much higher levels of IFN-I than innate cells.

Recently, it has been reported that human T cells transduced with a STING mutant from patients carrying the constitutively active mutation showed reduced proliferation and the patients showed an altered proportion of peripheral T-cell compartments characterized by increased naive T cells and reduced memory-type T-cell populations (Cerboni et al, 2017). Interestingly, T cell–specific Raptor-KO mice also showed reduced memory-type T-cell populations (Yang et al, 2013), suggesting the possibility that the activation of mTORC1 in these patients with activating STING mutations may be impaired.

In summary, our data demonstrate that T cell–intrinsic STING signaling and TCR signaling are coregulated to modulate T-cell functions. Our study has identified a key role of mTORC1-mediated signaling for both STING-mediated growth inhibition and IFN-I responses. These observations could have implications for the development of new therapeutic strategies for cancer, infectious diseases, and autoimmune diseases.

# Materials and Methods

## Mice

C57BL/6 mice were purchased from Clea Japan, Inc. Mice deficient in Tbk1, Tnf, and Ikk-ε were kindly provided by S Akira (Osaka University). IRF3-KO, IRF7-KO, and IRF3/7-DKO mice were provided by T Taniguchi (Tokyo University). $Rip3^{-/-}$ mice were kindly provided by Genentech, Inc. $Sting^{-/-}$ and $Ifnar1^{-/-}$ mice were kindly provided by GN Barber (University of Miami) and K Miyake (Tokyo University), respectively. $Rptor^{fl/fl}$ mice crossed with Lck-Cre mice were kindly provided by S Matsuda (Kansai Medical University). 6–16-wk-old mice were used. All mice were maintained under specific pathogen–free conditions at RIKEN, and all experiments were conducted under protocols approved by RIKEN Yokohama Institute.

## Cell preparation

CD4$^+$ and CD8$^+$ naive T cells were purified from spleen and lymph nodes as CD4$^+$/CD25$^-$/NK1.1$^-$/CD44$^{low}$/CD62L$^{high}$ cells by sorting using FACSAria (BD Biosciences). Th1 effector cells were prepared by stimulation of CD4$^+$ T cells with anti-CD3/CD28 Abs and cultured in the presence of IL-2 (10 ng/ml), IL-12 (10 ng/ml), and anti-IL-4 Ab (10 ng/ml) for 6 d in RPMI1640 medium supplemented with 10% FCS. Activated CD8$^+$ T cells were prepared by stimulating CD8$^+$/CD25$^-$/NK1.1$^-$/CD44$^{low}$/CD62L$^{high}$ (naive) T cells sorted by FACSAria with plate-bound anti-CD3ε (2C11, 10 µg/ml) and anti-CD28 (PV-1, 10 µg/ml) (anti-CD3/CD28) and then cultured in the presence of IL-2 (10 ng/ml).

B cells were prepared by sorting B220$^+$ cells from splenocytes. BMDCs were prepared by culturing bone marrow cells in the presence of IL-3 and sorted for CD11c$^+$ cells by FACSAria.

## Functional analyses

T cells were stimulated with immobilized anti-CD3ε (2C11, 10 µg/ml) and anti-CD28 (PV-1, 10 µg/ml) Ab with or without STING ligands. For antigen-specific activation of T cells, CD4$^+$ T cells from OVA-specific TCR-Tg mice OT-II were stimulated by coculturing with T cell–depleted splenocytes as APCs in the presence of OVA$^{323-339}$ peptide. Culture supernatants from these cultures were analyzed by ELISA for production of IL-2 (BD Biosciences), IFN-α (PBL assay science), IFN-β (PBL assay science), and IFN-λ2/3 (PBL assay science). Cell growth was assessed using a Cell Counting Kit-8 (DOJINDO). For apoptosis analysis, cells were stained with propidium iodide and annexin V and analyzed by FACS.

## Real-time qPCR

After removal of genomic DNA by treatment with DNase (Wako Nippon Gene), randomly primed cDNA strands were generated with reverse transcriptase III (Invitrogen). RNA expression was quantified by real-time PCR with gene-specific primers, and the values were normalized to the expression of $Rps18$ mRNA. qPCR was performed with the Fast SYBR Green Master Mix (Applied Biosystems). Data were collected and calculated by using the StepOnePlus real-time PCR system (Applied Biosystems).

## Reagents and Abs

The STING ligands cGAMP and c-di-AMP were purchased from Invivogen. DMXAA and etoposide were obtained from Sigma-Aldrich. Z-VAD-FMK and Rapamycin were obtained from Calbiochem.

Abs specific for anti-cyclin A (C-19, 1:1,000 dilution), anti-cyclin B1 (H-433, 1:1,000 dilution), anti-cyclin E (M-20, 1:1,000 dilution), anti-Cdk1 (17, 1:1,000 dilution), and anti-Cdk2 (H298, 1:1,000 dilution); anti-phospho-S6K1 (#9205, 1:1,000 dilution), anti-phospho-S6 (#2211, 1:1,000 dilution), anti-phospho-4E-BP1 (#9459, 1:1,000), anti-phospho-Akt (#9271, 1:1,000 dilution), anti-phospho-IRF3 (#4947, 1:1,000 dilution), anti-phospho-STAT5 (#9351, 1:1,000 dilution), anti-phospho JAK3 (#5031, 1:1,000 dilution), anti-STING (#13647, 1:1,000 dilution), anti-phospho-TBK1 (#5483, 1:1,000 dilution), anti-ERK (#9102, 1:1,000 dilution), anti-cleaved PARP (#9548, 1:1,000 dilution), anti-cleaved Caspase-3 (#9661, 1:1,000 dilution), anti-TBK1 (#3013, 1:1,000 dilution), anti-IKKε (#2690, 1:1,000 dilution), and anti-IRF3 (#4302, 1:1,000 dilution) were obtained from Cell Signaling Technology; anti-IRF7 (EPR4718, 1:1,000 dilution) was obtained from Abcam; anti-CD98 FITC (10.3, 1:50 dilution) was obtained from MBL. Flow cytometric analysis was performed on a FACSCalibur or LSR Fortessa X-20 and data were analyzed with CellQuest Pro or FlowJo.

## Western blot analysis

Cells were lysed in 1% Nonidet P-40 (NP-40) lysis buffer (1% NP-40, 50 mM Tris, 150 mM NaCl, 5 mM EDTA, 10 µg/ml of aprotinin, 12.5 µg/ml of chymostatin, 50 µg/ml of leupeptin, 25 µg/ml of

pepstatin A, 1 mM phenylmethylsulfonyl fluoride, and 2 mM $Na_3VO_4$). The lysates or immunoprecipitates were subjected by SDS–PAGE, and Western blots were carried out for the transferred membrane by reacting with specific Ab and developed with an enhanced chemiluminescence assay according to the manufacturer's recommendations (Pierce).

### Retroviral transduction

4E-BP1 (T37/46 AA) was cloned into the retroviral vector pMIG (provided by T. Kitamura, University of Tokyo). The construct was transiently transduced into Phoenix packaging cells (provided by G Nolan, Stanford University) using Lipofectamine with PLUS reagent (Invitrogen). Naive CD4[+] T cells were stimulated with plate-bound anti-CD3/CD28 Abs, and the cells were transduced by centrifugation at 1,640$g$ for 120 min in retroviral supernatants plus 8 $\mu$g/ml of polybrene (Sigma-Aldrich) on day 1 after stimulation. After 72 h of stimulation, the cells were sorted with a FACSAria to obtain GFP-positive populations.

### RNA-seq analysis

Total RNA was isolated from T cells using Direct-zol RNA kits (ZYMO RESEARCH) according to the manufacturer's instructions. The DNA library for RNA-seq analysis was constructed with NEBNext Ultra RNA Library Prep Kit for Illumina (NEB Biolabs, Inc) according to the manufacturer's instruction. The size range of the resulting DNA library was estimated on a 2100 Bioanalyzer (Agilent Technologies). The DNA library was subjected to the HiSeq 1500 sequencing system (Illumina) in a single-end read mode to obtain the sequencing data. The sequence reads were mapped to the *Mus musculus* reference genome (NCBI version 37) using TopHat2 version 2.0.8 and botwie2 version 2.1.0 with default parameters, and gene annotation was provided by NCBI. According to the mapped data, Cufflinks (version 2.1.1) was used to calculate the FPKM (fragments per kilobase per million mapped reads) values. Pathway-enrichment analysis was performed using DAVID Bioinformatics Resources 6.8 (Huang da et al, 2009). Heat maps were produced from normalized expression data referring to DAVID Bioinformatics Resources 6.8.

### Lipid analysis

Lipid extraction of CD4[+] T cells was performed as described (Tsugawa et al, 2017). Briefly, chloroform (100 $\mu$l) was added to dried cells in a tube, followed by 30 s sonication. After 60-min incubation at room temperature, 200 $\mu$l of MeOH was added and vortexed for 10 s. After 120 min of incubation, 20 $\mu$l of Milli-Q water was added, vortexed again, and the tube was left to stand for 10 min. The tubes were then centrifuged at 2,000$g$ for 10 min at 20°C, and the supernatant was transferred to LC-MS vials.

LC-MS/MS and LC-MS were used for identification and quantification of lipids. Non-targeted lipidomics analysis was performed as described (Takatani et al, 2015; Hirabayashi et al, 2017). Briefly, dried total lipid extracts were redissolved in 50 ml of chloroform: methanol (2:1, vol/vol), and 2 ml of samples were separated by an ACQUITY UPLC BEH C18 column (50_2.1 mm i.d., particle size 1.7 $\mu$m, Waters) at a flow rate of 300 $\mu$l/min at 45°C using an ACQUITY UPLC

system (Waters) equipped with a binary pump and automatic sample injector. Solvent A consisted of acetonitrile/methanol/water (20:20:60, vol/vol/vol) and solvent B was isopropanol, both containing 5 mM ammonium acetate. The solvent composition started at 100% A for the first 1 min and was changed linearly to 64% B at 7.5 min, where it was held for 4.5 min. The gradient was increased linearly to 82.5% B at 12.5 min, followed by 85% B at 19 min and 95% B at 20 min before re-equilibrating the column with 100% A for 5 min. Qualitative and quantitative analysis of lipids was performed by MS and data-dependent MS/MS acquisition with a scan range of $m/z$ 70–1,250 using a Triple TOF 5600[+] System (AB SCIEX) in the negative and positive ion mode. Raw data files from the TOF-MS were converted to MGF files using the program AB SCIEX MS converter for subsequent quantitative analysis with 2DICAL (Mitsui Knowledge Industry). Identification of molecular species was accomplished by comparison with retention times and MS/MS spectra with commercially available standards or reference samples.

### In vivo tumor growth and treatment

B16 F10 melanoma cells were cultured in complete DMEM media supplemented 10% heat-inactivated FBS. RAG1-KO mice were reconstituted with 2 × 10[6] *Sting*[+/−] or *Sting*[−/−] T cells and 4 × 10[6] WT B cells. After more than 30 d after reconstitution, T cell–specific STING-KO mice and control mice were injected s.c. on the back with a total of 2 × 10[5] B16 F10 cells on day 0. On days 8, 10, and 13, the mice were given intratumor injections of cGAMP (10 $\mu$g), and the mice were monitored for tumor growth and mortality. The tumor area was measured with a digital caliper and calculated using the formula: largest diameter × smallest diameter.

### Statistics

Statistical significance was determined by a two-tailed unpaired $t$ test. $P < 0.05$ was considered statistically significant.

## Data Availability

RNA sequencing data can be found publicly available on the National Center for Biotechnology Information (NCBI) Gene Expression Omnibus (GEO) website under the accession number for GSE104725.

## Supplementary Information

## Acknowledgements

We thank T Taniguchi for providing IRF3-KO, IRF7-KO, and IRF3/7-DKO mice; T Yokosuka, A Takeuchi, S Tsukumo, R Onishi, A Hashimoto-Tane, M Badr, N Hayatsu, H Ishigame, H Negishi, K Miyauchi, M Kubo, and H Hara for discussions and experimental help; M Sakuma for technical support; P Burrows for helpful comments on the manuscript; and M Yoshioka for secretarial

assistance. This work was supported by a Grant-in-Aid for Scientific Research from the Japan Society for the Promotion of Science KAKENHI (grants 16K08852 for T Imanishi and 24229004 for T Saito).

## Author Contributions

T Imanishi: conceptualization, data curation, formal analysis, funding acquisition, validation, investigation, and writing—original draft and project administration.
M Unno: investigation and data curation.
W Kobayashi: investigation and data curation.
N Yoneda: investigation and data curation.
S Matsuda: investigation and resources.
K Ikeda: investigation and data curation.
T Hoshii: resources.
A Hirao: resources.
K Miyake: resources.
GN Barber: resources.
M Arita: investigation and data curation.
KJ Ishii: resources.
S Akira: resources.
T Saito: conceptualization, supervision, funding acquisition, validation, project administration, and writing—review and editing.

## Conflict of Interest Statement

The authors declare that they have no conflict of interest.

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
