## [Reviewer comments · Life Science Alliance]

Life Science Alliance

Reciprocal regulation of STING and TCR signaling by mTORC1 for T cell activation and function

Takayuki Imanishi, Midori Unno, Wakana Kobayashi, Natsumi Yoneda, Satoshi Matsuda, Kazutaka Ikeda, Takayuki Hoshii, Atsushi Hirao, Kensuke Miyake, Glen Barber, Makoto Arita, Ken Ishii, Shizuo Akira, and Takashi Saito

DOI: [10.26508/lsa.201800282](https://doi.org/10.26508/lsa.201800282)

Corresponding author(s): Takashi Saito, RIKEN Center for Integrative Medical Sciences and Takayuki Imanishi, RIKEN Center for Integrative Medical Sciences (IMS)

Review Timeline:	Submission Date:	2018-12-17
	Revision Received:	2018-12-17
	Editorial Decision:	2018-12-17
	Revision Received:	2018-12-26
	Accepted:	2019-01-07

Scientific Editor: Andrea Leibfried

Transaction Report:

Please note that the manuscript was previously reviewed at another journal and the reports were taken into account in the decision-making process at Life Science Alliance. Since the original reviews are not subject to Life Science Alliance's transparent review process policy, the reports and author response cannot be published

December 17, 2018

RE: Life Science Alliance Manuscript #LSA-2018-00282-T

Prof. Takashi Saito
RIKEN Center for Integrative Medical Sciences
Laboratory for Cell Signaling
1-7-22 Suehiro-cho Tsurumi-ku
Yokohama, Kanagawa 230-0045
Japan

Dear Dr. Saito,

Thank you for transferring your revised manuscript entitled "Reciprocal regulation of STING and TCR signaling by mTORC1 for T cell activation and functions" to Life Science Alliance. Your work was reviewed at another journal twice before, and those reviewer reports were transferred to us with your permission.

One of the reviewers who re-reviewed your work elsewhere before was satisfied with the revision already performed, while another reviewer thought that the work did not provide sufficient mechanistic insight, especially not into whether STING-mediated IFN production inhibits the mTOR pathway via IFNAR. This lack of in-depth insight is not a concern for publication here, and I would therefore like to invite you to provide a final version of your manuscript. This revised version should address the comments made by reviewer #3 on STING departure from the ER and on the effect of cell cycling on cGAMP abundance in the cytosol as well as the lack of mechanistic insight mentioned above via text changes. I think re-writing the text slightly will also help others to better understand your findings. I would like to furthermore encourage a model figure as suggested by reviewer #1, a better description of rapamycin treatments and effects (Figure S5A description seems not accurate of the effects shown), as well as an acknowledgment that the exact mTORC1 activity regulation (on/off in the same context) needs further investigation.

The following editorial points should get addressed in the final manuscript version, too:

- please make sure to enter all necessary information in our submission system, including author contributions, a running title, summary blurb, and subject categories
- please note that author TM is listed as one of the authors in your author contribution section, but is not present on the author list, while Takayuki Hoshi is not listed in the author contribution section
- please note that Fig2E and SFig5B are not called out in the manuscript text
- please note that you mention Fig2F in your manuscript text, but the figure has no panel 'F'
- please note that SFig5G is not mentioned in the figure legend
- please make sure that all corresponding authors link an ORCID iD to their profile in our system

To avoid unnecessary delays in the acceptance and publication of your paper, please read the

following information carefully.

A. FINAL FILES:

-- High-resolution figure, supplementary figure and video files uploaded as individual files: See our detailed guidelines for preparing your production-ready images, <http://life-science-alliance.org/authorguide>

B. MANUSCRIPT ORGANIZATION AND FORMATTING:

Full guidelines are available on our Instructions for Authors page, <http://life-science-alliance.org/authorguide>

Thank you for your attention to these final processing requirements.

December 26, 2018

We are responding to your requests for the revision as follows as point-by-point responses.
This revised version should address the comments made by reviewer #3 on STING departure from the ER

We also analyzed whether cGAMP treatment may affect ER stress pathways because it has been shown that STING departure from ER causes ER stress that inhibits mTOR pathway. However, phosphorylation of PERK, IRE1 α and eIF2 α , which represent ER stress transducers, was not altered by cGAMP treatment (Figure S5H), indicating that inhibition of mTOR was not induced by ER stress but by STING signals in T cells. We described in p12, line25-30 in the text, Fig. S5H as the figure.

and on the effect of cell cycling on cGAMP abundance in the cytosol as well as the lack of mechanistic insight mentioned above via text changes.

It is noted that sustained activation of IRF3 induced by cGAMP and TCR stimulation was observed as early as 15 hours after stimulation (Figure 3B) when T cell division was not yet induced, indicating that TCR-induced sustained phosphorylation of IRF3 is induced independently of cell division. We described in p10, line7-11 in the text.

I would like to furthermore encourage a model figure as suggested by reviewer #1, Figure S7. Schematic model of STING pathways to induce growth inhibition and IFN-I responses in T cells. Costimulation of STING by cGAMP and TCR induces the growth inhibition by partially blocking the mTORC1 activation, and induces the sustained activation of IRF3 and the partial activation of mTORC1, leading to IFN-I production. This IFN-I also induces the growth inhibition without affecting mTORC1 signals. We described in p33, line32-p34, line4 in the text, Fig. S7 as the figure.

a better description of rapamycin treatments and effects (Figure S5A description seems not accurate of the effects shown),

It is noted that the inhibition of T cell growth by rapamycin treatment was similarly induced to the simultaneous treatment with both rapamycin and anti-IFNAR1 blocking Ab (Fig. S5A) because

rapamycin treatment inhibited both mTORC1 activation and cGAMP-induced IFN-I production. We described in p13, line23-26 in the text.

as well as an acknowledgment that the exact mTORC1 activity regulation (on/off in the same context) needs further investigation.

Together, these findings indicate that activation of STING induces the partial inhibition of mTORC1 activation which is sufficient to inhibit T cell proliferation on the one hand whereas this partially remained activation of mTORC1 signals is critical for IFN-I production on the other hand (Fig. S6H). We described at p14, line24-28 in the text.

The following editorial points should get addressed in the final manuscript version, too:

- please make sure to enter all necessary information in our submission system, including author contributions, a running title, summary blurb, and subject categories.

We added a running title, summary blurb and subject categories in the front page p3 in the text.

- please note that author TM is listed as one of the authors in your author contribution section, but is not present on the author list, while Takayuki Hoshi is not listed in the author contribution section

We corrected the authors both in the contribution section and author list in p20, line13 in the text.

- please note that Fig2E and SFig5B are not called out in the manuscript text

We corrected these figures Fig2E_p9, line9 and SFig5B_p12, line8 to be appeared in the text.

- please note that you mention Fig2F in your manuscript text, but the figure has no panel 'F'
Since there is no F, we deleted the description.

- please note that SFig5G is not mentioned in the figure legend

We added the figure legend for Fig.S5G and S5H in p32 in the text.

- please make sure that all corresponding authors link an ORCID iD to their profile in our system
We have linked ORCID iD.

January 7, 2019

RE: Life Science Alliance Manuscript #LSA-2018-00282-TR

Prof. Takashi Saito
RIKEN Center for Integrative Medical Sciences
Laboratory for Cell Signaling
1-7-22 Suehiro-cho Tsurumi-ku
Yokohama, Kanagawa 230-0045
Japan

Dear Dr. Saito,

Thank you for submitting your Research Article entitled "Reciprocal regulation of STING and TCR signaling by mTORC1 for T cell activation and function". I appreciate the introduced changes and while future work will further dissect the mechanisms underlying your observations, it is a pleasure to let you know that your present manuscript is now accepted for publication in Life Science Alliance. Congratulations on this interesting work.

*****IMPORTANT:** If you will be unreachable at any time, please provide us with the email address of an alternate author. Failure to respond to routine queries may lead to unavoidable delays in publication.*******

DISTRIBUTION OF MATERIALS:

Again, congratulations on a very nice paper. I hope you found the review process to be constructive and are pleased with how the manuscript was handled editorially. We look forward to future exciting

submissions from your lab.

Sincerely,
